# Learning with Group Invariant Features:
# A Kernel Perspective.

**Youssef Mroueh**
IBM Watson Group
mroueh@us.ibm.com

**Stephen Voinea**[*]
CBMM, MIT.
voinea@mit.edu

**Tomaso Poggio**
CBMM, MIT .
tp@ai.mit.edu

[*]*Co-first author*

## Abstract

We analyze in this paper a random feature map based on a theory of invariance (*I-theory*) introduced in [1]. More specifically, a group invariant signal signature is obtained through cumulative distributions of group-transformed random projections. Our analysis bridges invariant feature learning with kernel methods, as we show that this feature map defines an expected Haar-integration kernel that is invariant to the specified group action. We show how this non-linear random feature map approximates this group invariant kernel uniformly on a set of $N$ points. Moreover, we show that it defines a function space that is dense in the equivalent Invariant Reproducing Kernel Hilbert Space. Finally, we quantify error rates of the convergence of the empirical risk minimization, as well as the reduction in the sample complexity of a learning algorithm using such an invariant representation for signal classification, in a classical supervised learning setting.

## 1  Introduction

Encoding signals or building similarity kernels that are invariant to the action of a group is a key problem in unsupervised learning, as it reduces the complexity of the learning task and mimics how our brain represents information invariantly to symmetries and various nuisance factors (change in lighting in image classification and pitch variation in speech recognition) [1, 2, 3, 4]. Convolutional neural networks [5, 6] achieve state of the art performance in many computer vision and speech recognition tasks, but require a large amount of labeled examples as well as augmented data, where we reflect symmetries of the world through virtual examples [7, 8] obtained by applying identity-preserving transformations such as shearing, rotation, translation, etc., to the training data. In this work, we adopt the approach of [1], where the representation of the signal is designed to reflect the invariant properties and model the world symmetries with group actions. The ultimate aim is to bridge unsupervised learning of invariant representations with invariant kernel methods, where we can use tools from classical supervised learning to easily address the statistical consistency and sample complexity questions [9, 10]. Indeed, many invariant kernel methods and related invariant kernel networks have been proposed. We refer the reader to the related work section for a review (Section 5) and we start by showing how to accomplish this invariance through group-invariant Haar-integration kernels [11], and then show how random features derived from a memory-based theory of invariances introduced in [1] approximate such a kernel.

### 1.1  Group Invariant Kernels

We start by reviewing group-invariant Haar-integration kernels introduced in [11], and their use in a binary classification problem. This section highlights the conceptual advantages of such kernels as well as their practical inconvenience, putting into perspective the advantage of approximating them with explicit and invariant random feature maps.

**Invariant Haar-Integration Kernels.** We consider a subset $\mathcal{X}$ of the hypersphere in $d$ dimensions $\mathbb{S}^{d-1}$. Let $\rho_{\mathcal{X}}$ be a measure on $\mathcal{X}$. Consider a kernel $k_0$ on $\mathcal{X}$, such as a radial basis function kernel. Let $G$ be a group acting on $\mathcal{X}$, with a normalized Haar measure $\mu$. $G$ is assumed to be a compact and unitary group. Define an invariant kernel $\mathcal{K}$ between $x, z \in \mathcal{X}$ through Haar-integration [11] as follows:

$$\mathcal{K}(x,z) = \int_G \int_G k_0(gx, g'z)d\mu(g)d\mu(g'). \tag{1}$$

As we are integrating over the entire group, it is easy to see that: $\mathcal{K}(g'x, gz) = \mathcal{K}(x,z), \ \forall g, g' \in G, \forall x, z \in \mathcal{X}$. Hence the Haar-integration kernel is invariant to the group action. The symmetry of $\mathcal{K}$ is obvious. Moreover, if $k_0$ is a positive definite kernel, it follows that $\mathcal{K}$ is positive definite as well [11]. One can see the Haar-integration kernel framework as another form of data augmentation, since we have to produce group-transformed points in order to compute the kernel.

**Invariant Decision Boundary.** Turning now to a binary classification problem, we assume that we are given a labeled training set: $S = \{(x_i, y_i) \mid x_i \in \mathcal{X}, y_i \in \mathcal{Y} = \{\pm 1\}\}_{i=1}^N$. In order to learn a decision function $f : \mathcal{X} \to \mathcal{Y}$, we minimize the following empirical risk induced by an $L$-Lipschitz, convex loss function $V$, with $V'(0) < 0$ [12]: $\min_{f \in \mathcal{H}_{\mathcal{K}}} \hat{\mathcal{E}}_V(f) := \frac{1}{N} \sum_{i=1}^N V(y_i f(x_i))$, where we restrict $f$ to belong to a hypothesis class induced by the invariant kernel $\mathcal{K}$, the so called Reproducing Kernel Hilbert Space $\mathcal{H}_{\mathcal{K}}$. The representer theorem [13] shows that the solution of such a problem, or the optimal decision boundary $f_N^*$ has the following form: $f_N^*(x) = \sum_{i=1}^N \alpha_i^* \mathcal{K}(x, x_i)$. Since the kernel $\mathcal{K}$ is group-invariant it follows that : $f_N^*(gx) = \sum_{i=1}^N \alpha_i \mathcal{K}(gx, x_i) = \sum_{i=1}^N \alpha_i \mathcal{K}(x, x_i) = f_N^*(x), \ \forall g \in G$. Hence the the decision boundary $f^*$ is group-invariant as well, and we have: $f_N^*(gx) = f_N^*(x), \forall g \in G, \forall x \in \mathcal{X}$.

**Reduced Sample Complexity.** We have shown that a group-invariant kernel induces a group-invariant decision boundary, but how does this translate to the sample complexity of the learning algorithm? To answer this question, we will assume that the input set $\mathcal{X}$ has the following structure: $\mathcal{X} = \mathcal{X}_0 \cup \mathcal{G}\mathcal{X}_0, \ \mathcal{G}\mathcal{X}_0 = \{z | z = gx, x \in \mathcal{X}_0, g \in G/\{e\}\}$, where $e$ is the identity group element. This structure implies that for a function $f$ in the invariant RKHS $\mathcal{H}_{\mathcal{K}}$, we have:

$$\forall z \in \mathcal{G}\mathcal{X}_0, \exists\, x \in \mathcal{X}_0, \exists\, g \in G \text{ such that, } z = gx, \text{ and } f(z) = f(x).$$

Let $\rho_y(x) = \mathbb{P}(Y = y | x)$ be the label posteriors. We assume that $\rho_y(gx) = \rho_y(x), \forall g \in G$. This is a natural assumption since the label is unchanged given the group action. Assume that the set $\mathcal{X}$ is endowed with a measure $\rho_{\mathcal{X}}$ that is also group-invariant. Let $f$ be the group-invariant decision function and consider the expected risk induced by the loss $V$, $\mathcal{E}_V(f)$, defined as follows:

$$\mathcal{E}_V(f) = \int_{\mathcal{X}} \sum_{y \in \mathcal{Y}} V(yf(x))\rho_y(x)\rho_{\mathcal{X}}(x)dx, \tag{2}$$

$\mathcal{E}_V(f)$ is a proxy to the misclassification risk [12]. Using the invariant properties of the function class and the data distribution we have by invariance of $f$, $\rho_y$, and $\rho$:

$$
\begin{aligned}
\mathcal{E}_V(f) &= \int_{\mathcal{X}_0} \sum_{y \in \mathcal{Y}} V(yf(x))\rho_y(x)\rho_{\mathcal{X}}(x)dx + \int_{\mathcal{G}\mathcal{X}_0} \sum_{y \in \mathcal{Y}} V(yf(z))\rho_y(z)\rho_{\mathcal{X}}(z)dz \\
&= \int_G d\mu(g) \int_{\mathcal{X}_0} \sum_{y \in \mathcal{Y}} V(yf(gx))\rho_y(gx)\rho_{\mathcal{X}}(x)dx \\
&= \int_G d\mu(g) \int_{\mathcal{X}_0} \sum_{y \in \mathcal{Y}} V(yf(x))\rho_y(x)\rho_{\mathcal{X}}(x)dx \quad \text{(By invariance of } f, \rho_y, \text{ and } \rho) \\
&= \int_{\mathcal{X}_0} \sum_{y \in \mathcal{Y}} V(yf(x))\rho_y(x)\rho_{\mathcal{X}}(x)dx.
\end{aligned}
$$

Hence, given an invariant kernel to a group action that is identity preserving, it is sufficient to minimize the empirical risk on the core set $\mathcal{X}_0$, and it generalizes to samples in $\mathcal{G}\mathcal{X}_0$.

Let us imagine that $\mathcal{X}$ is finite with cardinality $|\mathcal{X}|$; the cardinality of the core set $\mathcal{X}_0$ is a small fraction of the cardinality of $\mathcal{X}$: $|\mathcal{X}_0| = \alpha|\mathcal{X}|$, where $0 < \alpha < 1$. Hence, when we sample training points from $\mathcal{X}_0$, the maximum size of the training set is $N = \alpha|\mathcal{X}| << |\mathcal{X}|$, yielding a reduction in the sample complexity.

## 1.2 Contributions

We have just reviewed the group-invariant Haar-integration kernel. In summary, a group-invariant kernel implies the existence of a decision function that is invariant to the group action, as well as a reduction in the sample complexity due to sampling training points from a reduced set, a.k.a the core set $\mathcal{X}_0$.

Kernel methods with Haar-integration kernels come at a very expensive computational price at both training and test time: computing the Kernel is computationally cumbersome as we have to integrate over the group and produce virtual examples by transforming points explicitly through the group action. Moreover, the training complexity of kernel methods scales cubically in the sample size. Those practical considerations make the usefulness of such kernels very limited.

The contributions of this paper are on three folds:

1. We first show that a non-linear random feature map $\Phi : \mathcal{X} \to \mathbb{R}^D$ derived from a memory-based theory of invariances introduced in [1] induces an expected group-invariant Haar-integration kernel $K$. For fixed points $x, z \in \mathcal{X}$, we have: $\mathbb{E}\langle \Phi(x), \Phi(z)\rangle = K(x,z)$, where $K$ satisfies: $K(gx, g'z) = K(x,z), \forall g, g' \in G, x, z \in \mathcal{X}$.

2. We show a Johnson-Lindenstrauss type result that holds uniformly on a set of $N$ points that assess the concentration of this random feature map around its expected induced kernel. For sufficiently large $D$, we have $\langle \Phi(x), \Phi(z)\rangle \approx K(x,z)$, uniformly on an $N$ points set.

3. We show that, with a linear model, an invariant decision function can be learned in this random feature space by sampling points from the core set $\mathcal{X}_0$ i.e: $f_N^*(x) \approx \langle w^*, \Phi(x)\rangle$ and generalizes to unseen points in $\mathcal{G}\mathcal{X}_0$, reducing the sample complexity. Moreover, we show that those features define a function space that approximates a dense subset of the invariant RKHS, and assess the error rates of the empirical risk minimization using such random features.

4. We demonstrate the validity of these claims on three datasets: text (artificial), vision (MNIST), and speech (TIDIGITS).

## 2   From Group Invariant Kernels to Feature Maps

In this paper we show that a random feature map based on I-theory [1]: $\Phi : \mathcal{X} \to \mathbb{R}^D$ approximates a group-invariant Haar-integration kernel $K$ having the form given in Equation (1):

$$\langle \Phi(x), \Phi(z)\rangle \approx K(x,z).$$

We start with some notation that will be useful for defining the feature map. Denote the cumulative distribution function of a random variable $X$ by,

$$F_X(\tau) = \mathbb{P}(X \leq \tau),$$

Fix $x \in \mathcal{X}$, Let $g \in G$ be a random variable drawn according to the normalized Haar measure $\mu$ and let $t$ be a random template whose distribution will be defined later. For $s > 0$, define the following truncated cumulative distribution function (CDF) of the dot product $\langle x, gt\rangle$:

$$\psi(x, t, \tau) = \mathbb{P}_g(\langle x, gt\rangle \leq \tau) = F_{\langle x, gt\rangle}(\tau), \ \tau \in [-s, s], x \in \mathcal{X},$$

Let $\varepsilon \in (0,1)$. We consider the following Gaussian vectors (sampling with rejection) for the templates $t$:

$$t = n \sim \mathcal{N}\left(0, \frac{1}{d}I_d\right), \text{ if } \|n\|_2^2 < 1 + \varepsilon, \ t = \perp \text{ else }.$$

The reason behind this sampling is to keep the range of $\langle x, gt\rangle$ under control: The squared norm $\|n\|_2^2$ will be bounded by $1 + \varepsilon$ with high probability by a classical concentration result (See proof of Theorem 1 for more details). The group being unitary and $x \in \mathbb{S}^{d-1}$, we know that : $|\langle x, gt\rangle| \leq \|n\|_2 < \sqrt{1 + \varepsilon} \leq 1 + \varepsilon$, for $\varepsilon \in (0,1)$.

**Remark 1.** *We can also consider templates $t$, drawn uniformly on the unit sphere $\mathbb{S}^{d-1}$. Uniform templates on the sphere can be drawn as follows:*

$$t = \frac{\nu}{\|\nu\|_2}, \ \nu \sim \mathcal{N}(0, I_d),$$

*since the norm of a gaussian vector is highly concentrated around its mean $\sqrt{d}$, we can use the gaussian sampling with rejection. Results proved for gaussian templates (with rejection) will hold true for templates drawn at uniform on the sphere with different constants.*

Define the following kernel function,

$$K_s(x, z) \;\; = \;\; \mathbb{E}_t \int_{-s}^{s} \psi(x, t, \tau)\psi(z, t, \tau)d\tau,$$

where $s$ will be fixed throughout the paper to be $s = 1 + \varepsilon$ since the gaussian sampling with rejection controls the dot product to be in that range.

Let $\bar{g} \in G$. As the group is closed, we have $\psi(t, \bar{g}x, \tau) = \int_G \mathbb{I}_{\langle g\bar{g}x, t\rangle \leq \tau} d\mu(g) = \int_G \mathbb{I}_{\langle gx, t\rangle \leq \tau} d\mu(g) = \psi(t, x, \tau)$ and hence $K(gx, g'z) = K(x, z)$, for all $g, g' \in G$. It is clear now that $K$ is a group-invariant kernel.

In order to approximate $K$, we sample $|G|$ elements uniformly and independently from the group $G$, i.e. $g_i, i = 1 \dots |G|$, and define the normalized empirical CDF :

$$\phi(x, t, \tau) = \frac{1}{|G|\sqrt{m}} \sum_{i=1}^{|G|} \mathbb{I}_{\langle g_i t, x\rangle \leq \tau}, \;\; -s \leq \tau \leq s.$$

We discretize the continuous threshold $\tau$ as follows:

$$\phi\left(x, t, \frac{sk}{n}\right) = \frac{\sqrt{s}}{\sqrt{nm}|G|} \sum_{i=1}^{|G|} \mathbb{I}_{\langle g_i t, x\rangle \leq \frac{s}{n}k}, \;\; -n \leq k \leq n.$$

We sample $m$ templates independently according to the Gaussian sampling with rejection, $t_j, j = 1 \dots m$. We are now ready to define the random feature map $\Phi$:

$$\Phi(x) = \left[\phi\left(x, t_j, \frac{sk}{n}\right)\right]_{j=1\dots m, k=-n\dots n} \in \mathbb{R}^{(2n+1)\times m}.$$

It is easy to see that:

$$\lim_{n\to\infty} \mathbb{E}_{t,g} \langle \Phi(x), \Phi(z)\rangle_{\mathbb{R}^{(2n+1)\times m}} = \lim_{n\to\infty} \mathbb{E}_{t,g} \sum_{j=1}^{m} \sum_{k=-n}^{n} \phi\left(x, t_j, \frac{sk}{n}\right) \phi\left(z, t_j, \frac{sk}{n}\right) = K_s(x, z).$$

In Section 3 we study the geometric information captured by this kernel by stating explicitly the similarity it computes.

**Remark 2** (Efficiency of the representation). *1) The main advantage of such a feature map, as outlined in [1], is that we store transformed templates in order to compute $\Phi$, while if we wanted to compute an invariant kernel of type $\mathcal{K}$ (Equation (1)), we would need to explicitly transform the points. The latter is computationally expensive. Storing transformed templates and computing the signature $\Phi$ is much more efficient. It falls in the category of memory-based learning, and is biologically plausible [1].*
*2) As $|G|, m, n$ get large enough, the feature map $\Phi$ approximates a group-invariant Kernel, as we will see in next section.*

## 3 An Equivalent Expected Kernel and a Uniform Concentration Result

In this section we present our main results, with proofs given in the supplementary material . Theorem 1 shows that the random feature map $\Phi$, defined in the previous section, corresponds in expectation to a group-invariant Haar-integration kernel $K_s(x, z)$. Moreover, $s - K_s(x, z)$ computes the average pairwise distance between all points in the orbits of $x$ and $z$, where the orbit is defined as the collection of all group-transformations of a given point $x : \mathcal{O}_x = \{gx, g \in G\}$.

**Theorem 1** (Expectation). *Let $\varepsilon \in (0, 1)$ and $x, z \in \mathcal{X}$. Define the distance $d_G$ between the orbits $\mathcal{O}_x$ and $\mathcal{O}_z$:*

$$d_G(x, z) = \frac{1}{\sqrt{2\pi d}} \int_G \int_G \|gx - g'z\|_2 \, d\mu(g)d\mu(g'),$$

*and the group-invariant expected kernel*

$$K_s(x, z) = \lim_{n\to\infty} \mathbb{E}_{t,g} \langle \Phi(x), \Phi(z)\rangle_{\mathbb{R}^{(2n+1)\times m}} = \mathbb{E}_t \int_{-s}^{s} \psi(x, t, \tau)\psi(z, t, \tau)d\tau, \; s = 1 + \varepsilon.$$

1. *The following inequality holds with probability 1:*

$$\varepsilon - \delta_2(d,\varepsilon) \le K_s(x,z) - (1 - d_G(x,z)) \le \varepsilon + \delta_1(d,\varepsilon), \tag{3}$$

   *where $\delta_1(\varepsilon,d) = \frac{e^{-d\varepsilon^2/16}}{\sqrt{d}} - \frac{1}{2}\frac{e^{-\varepsilon d/2}(1+\varepsilon)^{\frac{d}{2}}}{\sqrt{d}}$ and $\delta_2(\varepsilon,\delta) = \frac{e^{-d\varepsilon^2/16}}{\sqrt{d}} + (1+\varepsilon)e^{-d\varepsilon^2/8}$.*

2. *For any $\varepsilon \in (0,1)$ as the dimension $d \to \infty$ we have $\delta_1(\varepsilon,d) \to 0$ and $\delta_2(\varepsilon,d) \to 0$, and we have asymptotically $K_s(x,z) \to 1 - d_G(x,z) + \varepsilon = s - d_G(x,z)$.*

3. *$K_s$ is symmetric and $K_s$ is positive semi-definite.*

**Remark 3.** *1) $\varepsilon, \delta_1(d,\varepsilon)$, and $\delta_2(d,\varepsilon)$ are not errors due to results holding with high probability but are due to the truncation and are a technical artifact of the proof. 2) Local invariance can be defined by restricting the sampling of the group elements to a subset $\mathcal{G} \subset G$. Assuming that for each $g \in \mathcal{G}, g^{-1} \in \mathcal{G}$, the equivalent kernel has asymptotically the following form:*

$$K_s(x,z) \approx s - \frac{1}{\sqrt{2\pi d}} \int_{\mathcal{G}} \int_{\mathcal{G}} \|gx - g'z\|_2 \, d\mu(g)d\mu(g').$$

*3) The norm-one constraint can be relaxed, let $R = \sup_{x \in \mathcal{X}} \|x\|_2 < \infty$, hence we can set $s = R(1+\varepsilon)$, and*

$$-\delta_2(d,\varepsilon) \le K_s(x,z) - (R(1+\varepsilon) - d_G(x,z)) \le \delta_1(d,\varepsilon), \tag{4}$$

*where $\delta_1(\varepsilon,d) = R\frac{e^{-d\varepsilon^2/16}}{\sqrt{d}} - \frac{R}{2}\frac{e^{-\varepsilon d/2}(1+\varepsilon)^{\frac{d}{2}}}{\sqrt{d}}$ and $\delta_2(\varepsilon,\delta) = R\frac{e^{-d\varepsilon^2/16}}{\sqrt{d}} + R(1+\varepsilon)e^{-d\varepsilon^2/8}$.*

Theorem 2 is, in a sense, an invariant Johnson-Lindenstrauss [14] type result where we show that the dot product defined by the random feature map $\Phi$, i.e $\langle \Phi(x), \Phi(z) \rangle$, is concentrated around the invariant expected kernel uniformly on a data set of $N$ points, given a sufficiently large number of templates $m$, a large number of sampled group elements $|G|$, and a large bin number $n$. The error naturally decomposes to a numerical error $\varepsilon_0$ and statistical errors $\varepsilon_1, \varepsilon_2$ due to the sampling of the templates and the group elements respectively.

**Theorem 2.** *[Johnson-Lindenstrauss type Theorem- $N$ point Set] Let $\mathcal{D} = \{x_i \mid x_i \in \mathcal{X}\}_{i=1}^N$ be a finite dataset. Fix $\varepsilon_0, \varepsilon_1, \varepsilon_2, \delta_1, \delta_2 \in (0,1)$. For a number of bins $n \ge \frac{1}{\varepsilon_0}$, templates $m \ge \frac{C_1}{\varepsilon_1^2}\log(\frac{N}{\delta_1})$, and group elements $|G| \ge \frac{C_2}{\varepsilon_2^2}\log(\frac{Nm}{\delta_2})$, where $C_1, C_2$ are universal numeric constants, we have:*

$$|\langle \Phi(x_i), \Phi(x_j) \rangle - K_s(x_i, x_j)| \le \varepsilon_0 + \varepsilon_1 + \varepsilon_2, i = 1 \ldots N, j = 1 \ldots N, \tag{5}$$

*with probability $1 - \delta_1 - \delta_2$.*

Putting together Theorems 1 and 2, the following Corollary shows how the group-invariant random feature map $\Phi$ captures the invariant distance between points uniformly on a dataset of $N$ points.

**Corollary 1** (Invariant Features Maps and Distances between Orbits). *Let $\mathcal{D} = \{x_i \mid x_i \in \mathcal{X}\}_{i=1}^N$ be a finite dataset. Fix $\varepsilon_0, \delta \in (0,1)$. For a number of bins $n \ge \frac{3}{\varepsilon_0}$, templates $m \ge \frac{9C_1}{\varepsilon_0^2}\log(\frac{N}{\delta})$, and group elements $|G| \ge \frac{9C_2}{\varepsilon_0^2}\log(\frac{Nm}{\delta})$, where $C_1, C_2$ are universal numeric constants, we have:*

$$\varepsilon - \delta_2(d,\varepsilon) - \varepsilon_0 \le \langle \Phi(x_i), \Phi(x_j) \rangle - (1 - d_G(x_i, x_j)) \le \varepsilon_0 + \varepsilon + \delta_1(d,\varepsilon), \tag{6}$$

$i = 1 \ldots N, j = 1 \ldots N$, *with probability $1 - 2\delta$.*

**Remark 4.** *Assuming that the templates are unitary and drawn form a general distribution $p(t)$, the equivalent kernel has the following form:*

$$K_s(x,z) = \int_{\mathcal{G}} \int_{\mathcal{G}} d\mu(g)d\mu(g') \left( \int s - \max(\langle x, gt \rangle, \langle z, g't \rangle)p(t)dt \right).$$

*Indeed when we use the gaussian sampling with rejection for the templates, the integral $\int \max(\langle x, gt \rangle, \langle z, g't \rangle)p(t)dt$ is asymptotically proportional to $\left\| g^{-1}x - g'^{,-1}z \right\|_2$. It is interesting to consider different distributions that are domain-specific for the templates and assess the number of the templates needed to approximate such kernels. It is also interesting to find the optimal templates that achieve the minimum distortion in equation 6, in a data dependent way, but we will address these points in future work.*

## 4 Learning with Group Invariant Random Features

In this section, we show that learning a linear model in the invariant, random feature space, on a training set sampled from the reduced core set $\mathcal{X}_0$, has a low expected risk, and generalizes to unseen test points generated from the distribution on $\mathcal{X} = \mathcal{X}_0 \cup \mathcal{G}\mathcal{X}_0$. The architecture of the proof follows ideas from [15] and [16]. Recall that given an $L$-Lipschitz convex loss function $V$, our aim is to minimize the expected risk given in Equation (2). Denote the CDF by $\psi(x, t, \tau) = \mathbb{P}(\langle gt, x\rangle \leq \tau)$, and the empirical CDF by $\hat{\psi}(x, t, \tau) = \frac{1}{|G|} \sum_{i=1}^{|G|} \mathbb{1}_{\langle g_i t, x\rangle \leq \tau}$. Let $p(t)$ be the distribution of templates $t$. The RKHS defined by the invariant kernel $K_s$, $K_s(x, z) = \int \int_{-s}^{s} \psi(x, t, \tau)\psi(z, t, \tau)p(t)dtd\tau$ denoted $\mathcal{H}_{K_s}$, is the completion of the set of all finite linear combinations of the form:

$$f(x) = \sum_i \alpha_i K_s(x, x_i), x_i \in \mathcal{X}, \alpha_i \in \mathbb{R}. \tag{7}$$

Similarly to [16], we define the following infinite-dimensional function space:

$$\mathcal{F}_p = \left\{ f(x) = \int \int_{-s}^{s} w(t, \tau)\psi(x, t, \tau)dtd\tau \mid \sup_{\tau, t} \frac{|w(t, \tau)|}{p(t)} \leq C \right\}.$$

**Lemma 1.** $\mathcal{F}_p$ is dense in $\mathcal{H}_{K_s}$. For $f \in \mathcal{F}_p$ we have $\mathcal{E}_V(f) = \int_{\mathcal{X}_0} \sum_{y \in \mathcal{Y}} V(yf(x))\rho_y(x)d\rho_{\mathcal{X}}(x)$, where $\mathcal{X}_0$ is the reduced core set.

Since $\mathcal{F}_p$ is dense in $\mathcal{H}_{K_s}$, we can learn an invariant decision function in the space $\mathcal{F}_p$, instead of learning in $\mathcal{H}_{K_s}$. Let $\Psi(x) = \left[ \hat{\psi}\left(x, t_j, \frac{sk}{n}\right) \right]_{j=1...m, k=-n...n}$. $\Psi$, and $\Phi$ are equivalent up to constants. We will approximate the set $\mathcal{F}_p$ as follows:

$$\tilde{\mathcal{F}} = \left\{ f(x) = \langle w, \Psi(x)\rangle = \frac{s}{n} \sum_{j=1}^{m} \sum_{k=-n}^{n} w_{j,k} \hat{\psi}\left(x, t_j, \frac{sk}{n}\right), t_j \sim p, j = 1 \dots m \mid \|w\|_{\infty} \leq \frac{C}{m} \right\}.$$

Hence, we learn the invariant decision function via empirical risk minimization where we restrict the function to belong to $\tilde{\mathcal{F}}$, and the sampling in the training set is restricted to the core set $\mathcal{X}_0$. Note that with this function space we are regularizing for convenience the norm infinity of the weights but this can be relaxed in practice to a classical Tikhonov regularization.

**Theorem 3** (Learning with Group invariant features). *Let $S = \{(x_i, y_i) \mid x_i \in \mathcal{X}_0, y_i \in \mathcal{Y}, i = 1 \dots N\}$, a training set sampled from the core set $\mathcal{X}_0$. Let $f_N^* = \arg\min_{f \in \tilde{\mathcal{F}}} \hat{\mathcal{E}}_V(f) = \frac{1}{N} \sum_{i=1}^{N} V(y_i f(x_i))$. Fix $\delta > 0$, then*

$$\mathcal{E}_V(f_N^*) \leq \min_{f \in \mathcal{F}_p} \mathcal{E}_V(f) + 2\frac{1}{\sqrt{N}}\left( 4LsC + 2V(0) + LC\sqrt{\frac{1}{2}\log\left(\frac{1}{\delta}\right)} \right)$$

$$+ \frac{2sLC}{\sqrt{m}}\left( 1 + \sqrt{2\log\left(\frac{1}{\delta}\right)} \right) + L\left( \frac{2sC}{\sqrt{|G|}}\left( 1 + \sqrt{2\log\left(\frac{m}{\delta}\right)} \right) + \frac{2sC}{n} \right),$$

*with probability at least $1 - 3\delta$ on the training set and the choice of templates and group elements.*

The proof of Theorem 3 is given in Appendix B. Theorem 3 shows that learning a linear model in the invariant random feature space defined by $\Phi$ (or equivalently $\Psi$), has a low expected risk. More importantly, this risk is arbitrarily close to the optimal risk achieved in an infinite-dimensional class of functions, namely $\mathcal{F}_p$. The training set is sampled from the reduced core set $\mathcal{X}_0$, and invariant learning generalizes to unseen test points generated from the distribution on $\mathcal{X} = \mathcal{X}_0 \cup \mathcal{G}\mathcal{X}_0$, hence the reduction in the sample complexity. Recall that $\mathcal{F}_p$ is dense in the RKHS of the Haar-integration invariant Kernel, and so the expected risk achieved by a linear model in the invariant random feature space is not far from the one attainable in the invariant RKHS. Note that the error decomposes into two terms. The first, $O(\frac{1}{\sqrt{N}})$, is statistical and it depends on the training sample complexity $N$. The other is governed by the approximation error of functions $\mathcal{F}_p$, with functions in $\tilde{\mathcal{F}}$, and depends on the number of templates $m$, number of group elements sampled $|G|$, the number of bins $n$, and has the following form $O(\frac{1}{\sqrt{m}}) + O\left(\sqrt{\frac{\log m}{|G|}}\right) + \frac{1}{n}$.

# 5 Relation to Previous Work

We now put our contributions in perspective by outlining some of the previous work on invariant kernels and approximating kernels with random features.

**Approximating Kernels.** Several schemes have been proposed for approximating a non-linear kernel with an explicit non-linear feature map in conjunction with linear methods, such as the Nyström method [17] or random sampling techniques in the Fourier domain for translation-invariant kernels [15]. Our features fall under the random sampling techniques where, unlike previous work, we sample both projections and group elements to induce invariance with an integral representation. We note that the relation between random features and quadrature rules has been thoroughly studied in [18], where sharper bounds and error rates are derived, and can apply to our setting.

**Invariant Kernels.** We focused in this paper on Haar-integration kernels [11], since they have an integral representation and hence can be represented with random features [18]. Other invariant kernels have been proposed: In [19] authors introduce transformation invariant kernels, but unlike our general setting, the analysis is concerned with dilation invariance. In [20], multilayer arccosine kernels are built by composing kernels that have an integral representation, but does not explicitly induce invariance. More closely related to our work is [21], where kernel descriptors are built for visual recognition by introducing a kernel view of histogram of gradients that corresponds in our case to the cumulative distribution on the group variable. Explicit feature maps are obtained via kernel PCA, while our features are obtained via random sampling. Finally the convolutional kernel network of [22] builds a sequence of multilayer kernels that have an integral representation, by convolution, considering spatial neighborhoods in an image. Our future work will consider the composition of Haar-integration kernels, where the convolution is applied not only to the spatial variable but to the group variable akin to [2].

# 6 Numerical Evaluation

In this paper, and specifically in Theorems 2 and 3, we showed that the random, group-invariant feature map $\Phi$ captures the invariant distance between points, and that learning a linear model trained in the invariant, random feature space will generalize well to unseen test points. In this section, we validate these claims through three experiments. For the claims of Theorem 2, we will use a nearest neighbor classifier, while for Theorem 3, we will rely on the regularized least squares (RLS) classifier, one of the simplest algorithms for supervised learning. While our proofs focus on norm-infinity regularization, RLS corresponds to Tikhonov regularization with square loss. Specifically, for performing $T-$way classification on a batch of $N$ training points in $\mathbb{R}^d$, summarized in the data matrix $X \in \mathbb{R}^{N \times d}$ and label matrix $Y \in \mathbb{R}^{N \times T}$, RLS will perform the optimization, $\min_{W \in \mathbb{R}^{m \times T}} \left\{ \frac{1}{N} ||Y - \Phi(X)W||_F^2 + \lambda ||W||_F^2 \right\}$, where $|| \cdot ||_F$ is the Frobenius norm, $\lambda$ is the regularization parameter, and $\Phi$ is the feature map, which for the representation described in this paper will be a CDF pooling of the data projected onto group-transformed random templates. All RLS experiments in this paper were completed with the GURLS toolbox [23]. The three datasets we explore are:

$\mathbf{X_{perm}}$ (Figure 1): An artificial dataset consisting of all sequences of length 5 whose elements come from an alphabet of 8 characters. We want to learn a function which assigns a positive value to any sequence that contains a target set of characters (in our case, two of them) regardless of their position. Thus, the function label is globally invariant to permutation, and so we project our data onto all permuted versions of our random template sequences.

**MNIST** (Figure 2): We seek local invariance to translation and rotation, and so all random templates are translated by up to 3 pixels in all directions and rotated between -20 and 20 degrees.

**TIDIGITS** (Figure 3): We use a subset of TIDIGITS consisting of 326 speakers (men, women, children) reading the digits 0-9 in isolation, and so each datapoint is a waveform of a single word. We seek local invariance to pitch and speaking rate [25], and so all random templates are pitch shifted up and down by 400 cents and warped to play at half and double speed. The task is 10-way classification with one class-per-digit. See [24] for more detail.

**Acknowledgements:** Stephen Voinea acknowledges the support of a Nuance Foundation Grant. This work was also supported in part by the Center for Brains, Minds and Machines (CBMM), funded by NSF STC award CCF 1231216.

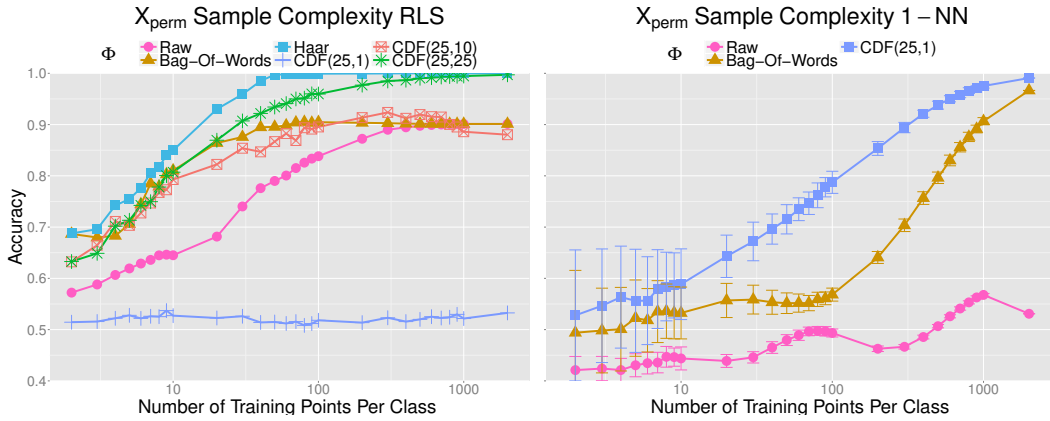

Figure 1: *Classification accuracy as a function of training set size, averaged over 100 random training samples at each size. $\Phi = CDF(n, m)$ refers to a random feature map with $n$ bins and $m$ templates. With 25 templates, the random feature map outperforms the raw features and a bag-of-words representation (also invariant to permutation) and even approaches an RLS classifier with a Haar-integration kernel. Error bars were removed from the RLS plot for clarity. See supplement.*

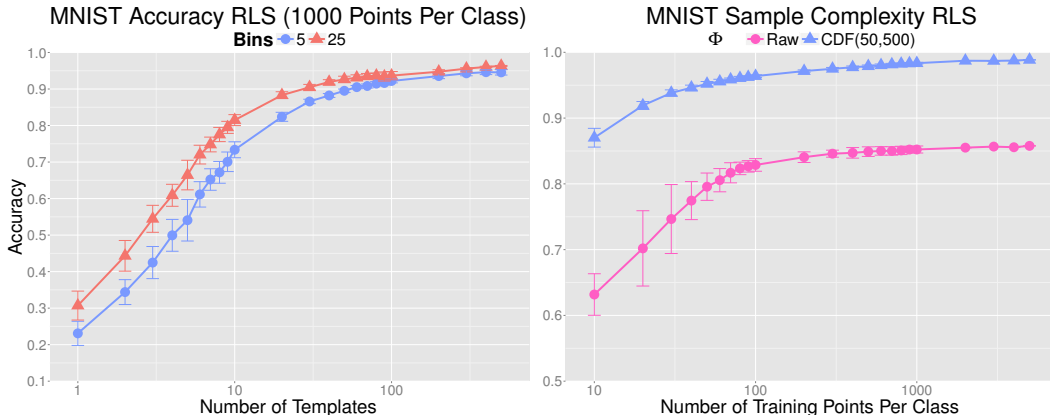

Figure 2: *Left Plot) Mean classification accuracy as a function of number of bins and templates, averaged over 30 random sets of templates. Right Plot) Classification accuracy as a function of training set size, averaged over 100 random samples of the training set at each size. At 1000 examples per class, we achieve an accuracy of 98.97%.*

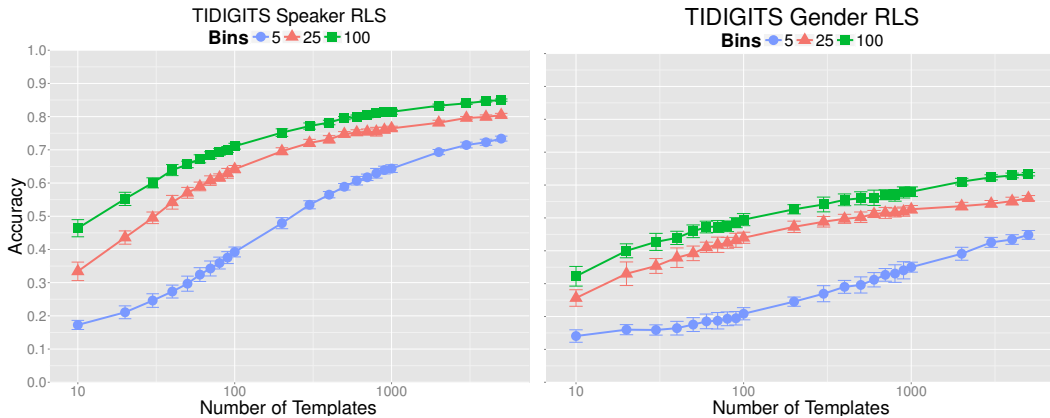

Figure 3: *Mean classification accuracy as a function of number of bins and templates, averaged over 30 random sets of templates. In the "Speaker" dataset, we test on unseen speakers, and in the "Gender" dataset, we test on a new gender, giving us an extreme train/test mismatch. [25].*

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
