[Supplementary Material]

# Supplementary Material.
# Learning with Group Invariant Features: A Kernel Perspective.

**Youssef Mroueh**
Multimodal Algorithms and Engines Group
IBM Watson,Yorktown Heights, NY 10598
mroueh@us.ibm.com

**Stephen Voinea   Tomaso Poggio**
Center for Brain Minds Machines.
MIT
voinea@mit.edu, tp@ai.mit.edu

## A   Proofs of Theorems 1 and 2

*Proof of Theorem 1.* 1)

$$
\begin{aligned}
K_s(x, z) &= \mathbb{E}_t \int_{-s}^{s} \mathbb{E}_g \left[ \mathbb{1}_{\langle x, gt \rangle \leq \tau} \right] \mathbb{E}_{g'} \left[ \mathbb{1}_{\langle z, g't \rangle \leq \tau} \right] d\tau \\
&= \mathbb{E}_t \int d\mu(g) d\mu(g') \int_{-s}^{s} \mathbb{1}_{\langle x, gt \rangle \leq \tau} \mathbb{1}_{\langle x, g't \rangle \leq \tau} d\tau \\
&= \int d\mu(g) d\mu(g') \mathbb{E}_t \left( s - \max(\langle x, gt \rangle, \langle z, g't \rangle) \right).
\end{aligned}
$$

where the second equality is by Fubini theorem and the last one holds since for $a, b \in [-s, s]$ :

$$
\int_{-s}^{s} \mathbb{1}_{a \leq \tau} \mathbb{1}_{b \leq \tau} d\tau = s - \max(a, b).
$$

Recall that the sampling of $t$ is the following for $\varepsilon \in (0, 1)$ let :

$$
t = n \sim \mathcal{N} \left( 0, \frac{1}{d} I_d \right), \text{ if } \|n\|_2^2 < 1 + \varepsilon, t = \perp \text{ else },
$$

since our group is unitary, $x$ being norm one, and by virtue of this sampling the dot product $|\langle x, gt \rangle| \leq \|n\|_2 \leq \sqrt{1+\varepsilon} \leq 1 + \varepsilon$ . Hence $\langle x, gt \rangle \in [-(1+\varepsilon), 1+\varepsilon]$, and we can choose $s = 1 + \varepsilon$. Using again the fact the group is unitary and compact we have:

$$
K_s(x, z) = \int d\mu(g) d\mu(g') \mathbb{E}_t (s - \max \left( \langle g^{-1} x, t \rangle, \langle g^{',-1} z, t \rangle \right)).
$$

Now using this particular sampling of templates we have:

$$
K_s(x, z) = \int_G \int_G d\mu(g) d\mu(g') \mathbb{E}_n \left( \mathbb{1}_{\|n\|_2^2 < 1+\varepsilon} \left[ 1 + \varepsilon - \max \left( \langle g^{-1} x, n \rangle, \langle g'^{-1} z, n \rangle \right) \right] \right).
$$

Let

$$
Z_{x,z}(n, g, g') = \max \left( \langle g^{-1} x, n \rangle, \langle g'^{-1} z, n \rangle \right),
$$

It follows that:

$$K_s(x,z) = \int_G \int_G d\mu(g)d\mu(g') \mathbb{E}_n \left( \mathbb{1}_{\|n\|_2^2 < 1+\varepsilon} \left[ 1 + \varepsilon - Z_{x,z}(n,g,g') \right] \right)$$

$$= (1+\varepsilon)\mathbb{P}(\|n\|_2^2 < 1+\varepsilon) - \int_G \int_G d\mu(g)d\mu(g')\mathbb{E}_n \left( \mathbb{1}_{\|n\|_2^2 < 1+\varepsilon} Z_{x,z}(n,g,g') \right)$$

$$= (1+\varepsilon)\mathbb{P}(\|n\|_2^2 < 1+\varepsilon) - \int_G \int_G d\mu(g)d\mu(g')\mathbb{E}_n \left( (1 - \mathbb{1}_{\|n\|_2^2 \geq 1+\varepsilon}) Z_{x,z}(n,g,g') \right)$$

$$= (1+\varepsilon)\mathbb{P}(\|n\|_2^2 < 1+\varepsilon) - \int_G \int_G d\mu(g)d\mu(g')\mathbb{E}_n Z_{x,z}(n,g,g')$$

$$+ \int_G \int_G d\mu(g)d\mu(g')\mathbb{E}_n \left( \mathbb{1}_{\|n\|_2^2 \geq 1+\varepsilon} Z_{x,z}(n,g,g') \right) \tag{1}$$

We are left with evaluating or bounding two expectations: $I_1 = \mathbb{E}_n Z_{x,z}(n,g,g')$, and $I_2 = \mathbb{E}_n \left( \mathbb{1}_{\|n\|_2^2 \geq 1+\varepsilon} Z_{x,z}(n,g,g') \right)$, that involve the maximum of correlated gaussian variables as we will see in the following.

By rotation invariance of Gaussians we have that $\langle g^{-1}x, n \rangle$, and $\langle g'^{-1}z, n \rangle$ are two correlated random gaussian variables with correllation coefficient that we note by $\cos(\theta_{g,g'}) = \langle g^{-1}x, g'^{-1}z \rangle$. Hence by a change of a basis we can write:

$$\langle g^{-1}x, n \rangle = \frac{1}{\sqrt{d}}u, \quad \langle g'^{-1}z, n \rangle = \frac{1}{\sqrt{d}}\cos(\theta_{g,g'})u + \frac{1}{\sqrt{d}}\sqrt{1 - \cos^2(\theta_{g,g'})}v$$

where $\cos(\theta_{g,g'}) = \langle g^{-1}x, g'^{-1}z \rangle$, and $u,v \sim \mathcal{N}(0,1)$ iids.

Hence,

$$I_1 = \frac{1}{\sqrt{d}}\mathbb{E}_{u,v} \max \left( u, \cos(\theta_{g,g'})u + \sqrt{1 - \cos^2(\theta_{g,g'})}v \right).$$

The following Lemma from [26] gives the expectation and the variance of the maximum of two gaussians with correllation coefficient $\rho$.

**Lemma 1** (Mean and Variance of Maximum of Correlated Gaussians [26] ). *Let* $X \sim \mathcal{N}(\mu_X, \sigma_X^2)$ *and* $Y \sim \mathcal{N}(\mu_Y, \sigma_Y^2)$, *two correlated gaussians with correllation coefficient* $\rho$. *Define* $\phi_{\mathcal{N}}(x) = \frac{1}{\sqrt{2\pi}}\exp(-x^2/2)$, *and* $\Phi_{\mathcal{N}}(y) = \int_{-\infty}^y \phi_{\mathcal{N}}(x)dx$. *Let* $a = \sqrt{\sigma_X^2 + \sigma_Y^2 - 2\rho\sigma_X\sigma_Y}$, *and* $\alpha = \frac{\mu_X - \mu_Y}{a}$.
*The mean* $\mu_Z$ *and variance* $\sigma_Z^2$ *of* $Z = \max(X,Y)$ *are expressed analytically as follows:*

$$\mu_Z = \mu_X \Phi_{\mathcal{N}}(\alpha) + \mu_Y \Phi_{\mathcal{N}}(-\alpha) + a\phi_{\mathcal{N}}(\alpha). \tag{2}$$

$$\sigma_Z^2 = \underbrace{\left(\sigma_X^2 + \mu_X^2\right)\Phi_{\mathcal{N}}(\alpha) + \left(\sigma_Y^2 + \mu_Y^2\right)\Phi_{\mathcal{N}}(-\alpha) + \left(\mu_X + \mu_Y\right)a\phi_{\mathcal{N}}(\alpha)}_{\mathbb{E}Z^2} - \mu_Z^2. \tag{3}$$

Applying Lemma 2 to our case ($\mu_X = \mu_Y = 0, \sigma_X = \sigma_Y = 1, \rho = \cos(\theta_{g,g'})$). We have: $a = \sqrt{2(1 - \cos(\theta_{g,g'}))}$ and $\alpha = 0$.

$$I_1 = \frac{1}{\sqrt{d}}a\phi_{\mathcal{N}}(0)$$

$$= \frac{1}{\sqrt{2\pi d}}\sqrt{2(1 - \cos(\theta_{g,g'}))}$$

$$= \frac{1}{\sqrt{2\pi d}}\left\|g^{-1}x - g'^{-1}z\right\|_2. \tag{4}$$

We turn now to $I_2$ that we bound using Cauchy-Schwarz inequality:

$$|I_2| = \left| \mathbb{E}_n \left( \mathbb{1}_{\|n\|_2^2 \geq 1+\varepsilon} Z_{x,z}(n,g,g') \right) \right|$$

$$\leq \sqrt{E(\mathbb{1}_{\|n\|_2^2 \geq 1+\varepsilon})}\sqrt{E(Z_{x,z}^2(n,g,g'))}$$

$$= \sqrt{\mathbb{P}\left(\|n\|_2^2 \geq 1 + \varepsilon\right)}\sqrt{E(Z_{x,z}^2(n,g,g'))}. \tag{5}$$

On the first hand, applying again Lemma 2 (for $\mathbb{E}Z^2$) we have:

$$
\begin{aligned}
E(Z_{x,z}^2(n,g,g')) &= \frac{1}{d}\mathbb{E}_{u,v}\left(\max\left(u,\cos(\theta_{g,g'})u+\sqrt{1-\cos^2(\theta_{g,g'})}v\right)\right)^2 \\
&= \frac{1}{d}\left(2\Phi_{\mathcal{N}}(0)\right) \\
&= \frac{1}{d}.
\end{aligned}
\tag{6}
$$

On the other hand, note that $\|n\|_2^2$ has a (normalized) chi squared distribution with $d$ degree of freedom $\chi_d^2$, with mean $1$. The following Lemma gives upper bounds for the upper and lower tails of a chi square distribution.

**Lemma 2** ($\chi^2$ tail bounds). *Let $X \sim \chi_k^2$, a chi squared random variable with $k$ degree of freedom. The following hold true for any $\varepsilon \in (0,1)$:*

- *Upper Bound for the upper tail [27]: $\mathbb{P}\left(\frac{1}{k}X \geq 1+\varepsilon\right) \leq e^{-k\varepsilon^2/8}$.*

- *Upper Bound for the lower tail [28]: For all $k \geq 2$, $u \geq k-1$ we have:*

$$
\mathbb{P}\left(X < u\right) \leq 1 - \frac{1}{2}\exp\left(-\frac{1}{2}\left(u-k-(k-2)\log(u/k)+\log(k)\right)\right).
$$

*More specifically for $u = k(1+\varepsilon)$ we have:*

$$
\mathbb{P}\left(\frac{1}{k}X < 1+\varepsilon\right) \leq 1 - \frac{1}{2}\frac{e^{-\varepsilon k/2}(1+\varepsilon)^{\frac{k-2}{2}}}{\sqrt{k}}.
$$

Applying Lemma 3, for $\|n\|_2^2$. We have $\|n\|_2^2 = \frac{1}{d}X$, where $X \sim \chi_d^2$, hence:

$$
\mathbb{P}\left(\|n\|_2^2 \geq 1+\varepsilon\right) \leq e^{-d\varepsilon^2/8},
\tag{7}
$$

Putting together Equations (14),(16), (15) we have finally:

$$
|I_2| \leq \frac{e^{-d\varepsilon^2/16}}{\sqrt{d}}.
\tag{8}
$$

Putting together Equations (10), (13), and (17), and using upper and lower bounds for $\mathbb{P}(\|n\|_2^2 < 1+\varepsilon)$ from Lemma 3:

$$
\begin{aligned}
K_s(x,z) &\leq (1+\varepsilon)\,\mathbb{P}(\|n\|_2^2 < 1+\varepsilon) - \frac{1}{\sqrt{2\pi d}}\int_G\int_G \|g^{-1}x-g'^{-1}z\|_2\,d\mu(g)d\mu(g') + \frac{e^{-d\varepsilon^2/16}}{\sqrt{d}} \\
&\leq (1+\varepsilon)\left(1-\frac{1}{2}\frac{e^{-\varepsilon d/2}(1+\varepsilon)^{\frac{d-2}{2}}}{\sqrt{d}}\right) - \frac{1}{\sqrt{2\pi d}}\int_G\int_G \|g^{-1}x-g'^{-1}z\|_2\,d\mu(g)d\mu(g') \\
&\quad + \frac{e^{-d\varepsilon^2/16}}{\sqrt{d}}.
\end{aligned}
$$

$$
\begin{aligned}
K_s(x,z) &\geq (1+\varepsilon)\mathbb{P}(\|n\|_2^2 < 1+\varepsilon) - \frac{1}{\sqrt{2\pi d}}\int_G\int_G \|g^{-1}x-g'^{-1}z\|_2\,d\mu(g)d\mu(g') - \frac{e^{-d\varepsilon^2/16}}{\sqrt{d}} \\
&\geq (1+\varepsilon)\left(1-e^{-d\varepsilon^2/8}\right) - \frac{1}{\sqrt{2\pi d}}\int_G\int_G \|g^{-1}x-g'^{-1}z\|_2\,d\mu(g)d\mu(g') - \frac{e^{-d\varepsilon^2/16}}{\sqrt{d}}.
\end{aligned}
$$

Noting by $d_G$ the integral and using that the group is compact and unitary:

$$
\begin{aligned}
d_G(x,z) &= \frac{1}{\sqrt{2\pi d}}\int_G\int_G \|g^{-1}x-g'^{-1}z\|_2\,d\mu(g)d\mu(g') \\
&= \frac{1}{\sqrt{2\pi d}}\int_G\int_G \|gx-g'z\|_2\,d\mu(g)d\mu(g').
\end{aligned}
$$

We finally have:

$$-\frac{e^{-d\varepsilon^2/16}}{\sqrt{d}}-(1+\varepsilon)e^{-d\varepsilon^2/8}+\varepsilon \le K_s(x,z)-(1-d_G(x,z)) \le \frac{e^{-d\varepsilon^2/16}}{\sqrt{d}}-\frac{1}{2}\frac{e^{-\varepsilon d/2}\,(1+\varepsilon)^{\frac{d}{2}}}{\sqrt{d}}+\varepsilon.$$
(9)

For any $\varepsilon \in (0,1)$, as the dimension $d \to \infty$, we have asymptotically:

$$K_s(x,z) \to 1 - d_G(x,z) + \varepsilon = s - d_G(x,z).$$

2) The symmetry of $K$ is obvious. Let $p(t)$ be the distribution of the templates $t$. Define the following weighted dot product: $\langle f(x,.,.), g(z,.,.)\rangle = \int_t p(t) \int_{-s}^s d\tau f(x,t,\tau) g(z,t,\tau)$. Recall that:

$$\begin{aligned}K_s(x,z) &= \int p(t)dt \int_{-s}^s \psi(x,t,\tau)\psi(z,t,\tau)d\tau \\ &= \langle \psi(x,.,.), \psi(z,.,.)\rangle.\end{aligned}$$

Hence $K$ is symmetric and positive semidefinite.

$\square$

*Proof of Theorem 2.* In the following we fix two points $x$ and $z$ in $\mathcal{X}$ and a random template $t$. Let $X_j = \int_{-s}^s \mathbb{P}(\langle gt_j, x\rangle \le \tau)\mathbb{P}(\langle gt_j, z\rangle \le \tau)d\tau$, we have $0 \le X_j \le 2s$, where $s = 1+\varepsilon$. Recall that $K_s(x,z) = \frac{1}{m}\mathbb{E}_t(\sum_{j=1}^m X_j)$. By Hoeffding's inequality we have:

$$\mathbb{P}_t\left\{\left|\frac{1}{m}\sum_{j=1}^m X_j - K_s(x,z)\right| > \epsilon\right\} \le 2\exp\left(\frac{-2m\epsilon^2}{(2s)^2}\right)$$

Turning now to the CDF $\psi(x,t,\tau) = \mathbb{P}(\langle gt, x\rangle \le \tau)$, and the empirical CDF $\hat{\psi}(x,t,\tau) = \frac{1}{|G|}\sum_{i=1}^{|G|} \mathbb{1}_{\langle g_i t, x\rangle \le \tau}$. By the theorem on convergence of the empirical CDF [29] (Theorem 4 given in Appendix D ) we have, for $\gamma > 0$:

$$\mathbb{P}_g\left\{\sup_\tau \left|\hat{\psi}(x,t,\tau) - \psi(x,t,\tau)\right| > \gamma\right\} \le 2\exp(-2|G|\gamma^2)$$

Hence we have $\forall \tau \in [-s,s]$:

$$\left|\hat{\psi}(x,t,\tau) - \psi(x,t,\tau)\right| \le \gamma \text{ and } \left|\hat{\psi}(x,t,\tau) - \psi(z,t,\tau)\right| \le \gamma$$

with a probability at least $1 - 4\exp(-2|G|\gamma^2)$.
Define $X = \int_{-s}^s \psi(x,t,\tau)\psi(z,t,\tau)d\tau$, $\hat{X} = \int_{-s}^s \hat{\psi}(x,t,\tau)\hat{\psi}(z,t,\tau)d\tau$, and $\tilde{X} = \frac{(2s)}{n}\sum_{k=-n}^n \hat{\psi}(x,t,\frac{ks}{n})\hat{\psi}(z,t,\frac{ks}{n})$, choose $0 < \gamma < 1$:

$$\begin{aligned}|\hat{X} - X| &= \left|\int_{-s}^s \left(\hat{\psi}(x,t,\tau)\hat{\psi}(z,t,\tau) - \psi(x,t,\tau)\psi(z,t,\tau)\right)d\tau\right| \\ &= \left|\int_{-s}^s \left(\hat{\psi}(x,t,\tau)-\psi(x,t,\tau)+\psi(x,t,\tau)\right)\left(\hat{\psi}(z,t,\tau)-\psi(z,t,\tau)+\psi(z,t,\tau)\right) - \psi(x,t,\tau)\psi(z,t,\tau)d\tau\right| \\ &\le (2\gamma+\gamma^2)2s \\ &\le 6s\gamma,\end{aligned}$$

with probability $1 - 4\exp(-2|G|\gamma^2)$. Define $X_j = \int_{-s}^s \psi(x,t_j,\tau)\psi(z,t_j,\tau)d\tau$, $\hat{X}_j = \int_{-s}^s \hat{\psi}(x,t_j,\tau)\hat{\psi}(z,t_j,\tau)d\tau$, and $\tilde{X}_j = \frac{(2s)}{n}\sum_{k=-n}^n \hat{\psi}(x,t_j,\frac{ks}{n})\hat{\psi}(z,t_j,\frac{ks}{n})$, Then for all $j = 1\ldots m$, we have

$$|\hat{X}_j - X_j| \le 6s\gamma$$

with probability $1 - 4m\exp(-2|G|\gamma^2) - 2\exp\left(\frac{-2m\epsilon^2}{(2s)^2}\right).$

Now we turn to the numerical approximation of the integra by a Riemann sum, we have for all $j = 1 \dots m$:

$$\left| \hat{X}_j - \tilde{X}_j \right| \leq \frac{s}{n}.$$

Hence the error decomposes in the following way:

$$|\langle \Phi(x), \Phi(z) \rangle - K_s(x,z)| = \left| \frac{1}{m} \sum_{j=1}^{m} \tilde{X}_j - K_s(x,z) \right|$$

$$= \left| \left( \frac{1}{m} \sum_{j=1}^{m} \tilde{X}_j - \frac{1}{m} \sum_{j=1}^{m} \hat{X}_j \right) + \left( \frac{1}{m} \sum_{j=1}^{m} \hat{X}_j - \frac{1}{m} \sum_{j=1}^{m} X_j \right) + \left( \frac{1}{m} \sum_{j=1}^{m} X_j - K_s(x,z) \right) \right|$$

$$\leq \underbrace{\left| \frac{1}{m} \sum_{j=1}^{m} \tilde{X}_j - \frac{1}{m} \sum_{j=1}^{m} \hat{X}_j \right|}_{\text{Numerical Binning Error}} + \underbrace{\left| \frac{1}{m} \sum_{j=1}^{m} \hat{X}_j - \frac{1}{m} \sum_{j=1}^{m} X_j \right|}_{\text{Group CDF Approximation Error}} + \underbrace{\left| \frac{1}{m} \sum_{j=1}^{m} X_j - K_s(x,z) \right|}_{\text{Templates Concentration Error}}$$

$$\leq \frac{s}{n} + 6s\gamma + \epsilon.$$

with probability $1 - 4m \exp(-2|G|\gamma^2) - 2 \exp\left( \frac{-2m\epsilon^2}{(2s)^2} \right)$. For this to hold on all pairs of points in a set of cardinality $N$ we have:

$$|\langle \Phi(x_i), \Phi(x_j) \rangle - K(x_i, x_j)| \leq \frac{s}{n} + 6s\gamma + \epsilon, i = 1 \dots N, j = 1 \dots N,$$

with probability $1 - 4mN(N-1) \exp(-2|G|\gamma^2) - 2N(N-1) \exp\left( \frac{-m\epsilon^2}{2(s)^2} \right)$.

Hence we have for numerical constants $C_1$, and $C_2$, $0 < \delta_1, \delta_2 < 1$, and $0 < \varepsilon_0, \varepsilon_1, \varepsilon_2 < 1$, for $n \geq \frac{s}{\varepsilon_0}, m \geq \frac{C_1}{\varepsilon_1^2} \log(\frac{N}{\delta_1}), |G| \geq \frac{C_2}{\varepsilon_2^2} \log(\frac{Nm}{\delta_2})$, :

$$|\langle \Phi(x_i), \Phi(x_j) \rangle - K_s(x_i, x_j)| \leq \varepsilon_0 + \varepsilon_1 + \varepsilon_2, i = 1 \dots N, j = 1 \dots N,$$

with probability $1 - \delta_1 - \delta_2$.

$\square$

# B   Proof of Theorem 3

*Proof of Lemma 1.* Our proof parallels similar proofs in [16]. Note that functions of the form (9) are dense in $\mathcal{H}_K$. $f(x) = \sum_i \alpha_i K_s(x, x_i) = \sum_i \alpha_i \int \int_{-s}^{s} \psi(x, t, \tau) \psi(x_i, t, \tau) p(t) dt d\tau$
$= \int \int_{-s}^{s} \left( p(t) \sum_i \alpha_i \psi(x_i, t, \tau) \right) \psi(x, t, \tau) dt d\tau$. Let $\beta(t, \tau) = p(t) \sum_i \alpha_i \psi(x_i, t, \tau)$, since $0 \leq \psi(x, t, \tau) \leq 1, \forall x, t, \tau$, we have $\frac{|\beta(t,\tau)|}{p(t)} \leq \sum_i |\alpha_i| < \infty$, since $\alpha_i$ are finite. Hence $f$ can be written in the form:

$$f(x) = \int \int_{-s}^{s} \beta(t, \tau) \psi(x, t, \tau) dt d\tau, \ \sup_{\tau, t} \frac{|\beta(t, \tau)|}{p(t)} < \infty,$$

and $f \in \mathcal{F}_p$. $\square$

In order to prove Theorem 3, we need some preliminary lemmas. The following Lemma assess the approximation of any function $f \in \mathcal{F}_p$, by a certain $\tilde{f} \in \tilde{\mathcal{F}}$.

**Lemma 3** ($\tilde{\mathcal{F}}$ Approximation of $\mathcal{F}_p$)**.** *Let $f$ be a function in $\mathcal{F}_p$. Then for $\delta_1, \delta_2 > 0$, there exists a function $\tilde{f} \in \tilde{\mathcal{F}}$ such that:*

$$\left\| \tilde{f} - f \right\|_{\mathcal{L}^2(\mathcal{X}, \rho_{\mathcal{X}})} \leq \frac{2sC}{\sqrt{m}} \left( 1 + \sqrt{2 \log \left( \frac{1}{\delta_1} \right)} \right) + \frac{2sC}{\sqrt{|G|}} \left( 1 + \sqrt{2 \log \left( \frac{m}{\delta_2} \right)} \right) + \frac{2sC}{n},$$

*with probability at least $1 - \delta_1 - \delta_2$.*

*Proof of Lemma 4.* Let $f \in \mathcal{F}_p$, $f(x) = \int \int_{-s}^{s} w(t,\tau)\psi(x,t,\tau)d\tau dt$.

Let $f_j(x) = \int_{-s}^{s} \frac{w(t_j,\tau)}{p(t_j)}\psi(x,t_j,\tau)d\tau$, $\hat{f}_j(x) = \int_{-s}^{s} \frac{w(t_j,\tau)}{p(t_j)}\hat{\psi}(x,t_j,\tau)d\tau$, and $\tilde{f}_j(x) = \frac{s}{n}\sum_{k=-n}^{n} \frac{w(t_j,\frac{ks}{n})}{p(t_j)}\hat{\psi}(x,t_j,\frac{ks}{n})$. We have the following: $\mathbb{E}_t(f_j) = f$, and $\frac{1}{m}\mathbb{E}_t(\sum_{j=1}^{m} f_j) = f$.
Consider the Hilbert space $\mathcal{L}^2(\mathcal{X},\rho_{\mathcal{X}})$, with dot product: $\langle f,g \rangle_{\mathcal{L}^2(\mathcal{X},\rho_{\mathcal{X}})} = \int_{\mathcal{X}} f(x)g(x)d\rho_{\mathcal{X}}(x)$.

Note that : $\int_{-s}^{s} g(\tau)d\tau \leq \sqrt{2s}\sqrt{\int_{-s}^{s} g^2(\tau)d\tau}$

$$||f_j||_{\mathcal{L}^2(\mathcal{X},\rho_{\mathcal{X}})} = \sqrt{\int_{\mathcal{X}} \left( \int_{-s}^{s} \frac{w(t_j,\tau)}{p(t_j)}\psi(x,t_j,\tau)d\tau \right)^2 d\rho_{\mathcal{X}}(x)} \leq (2sC),$$

Fix $\delta_1 > 0$, applying Lemma 7 we have therefore with probability $1 - \delta_1$:

$$\left\| \frac{1}{m}\sum_{j=1}^{m} f_j - f \right\|_{\mathcal{L}^2(\mathcal{X},\rho_{\mathcal{X}})} \leq \frac{2sC}{\sqrt{m}}\left(1 + \sqrt{2\log\left(\frac{1}{\delta_1}\right)}\right), \tag{10}$$

Now turn to:

$$\left\| \frac{1}{m}\sum_{j=1}^{m}(\hat{f}_j - f_j) \right\|_{\mathcal{L}^2(\mathcal{X},\rho_{\mathcal{X}})} \leq \frac{1}{m}\sum_{j=1}^{m}\left\| \hat{f}_j - f_j \right\|_{\mathcal{L}^2(\mathcal{X},\rho_{\mathcal{X}})},$$

$$
\begin{aligned}
\left\| \hat{f}_j - f_j \right\|^2_{\mathcal{L}^2(\mathcal{X},\rho_{\mathcal{X}})} &= \int_{\mathcal{X}} \left( \int_{-s}^{s} \frac{w(t_j,\tau)}{p(t_j)}(\psi(x,t_j,\tau) - \hat{\psi}(x,t_j,\tau))d\tau \right)^2 d\rho_{\mathcal{X}}(x) \\
&\leq 2s \int_{\mathcal{X}} \int_{-s}^{s} \frac{w^2(t_j,\tau)}{p^2(t_j)}(\psi(x,t_j,\tau) - \hat{\psi}(x,t_j,\tau))^2 d\tau d\rho_{\mathcal{X}}(x) \\
&\leq 2sC^2 \int_{\mathcal{X}} \int_{-s}^{s} (\hat{\psi}(x,t_j,\tau) - \psi(x,t_j,\tau))^2 d\tau d\rho_{\mathcal{X}}(x) \\
&= 2sC^2 \int_{-s}^{s} \int_{\mathcal{X}} (\hat{\psi}(x,t_j,\tau) - \psi(x,t_j,\tau))^2 d\rho_{\mathcal{X}}(x)d\tau \\
&= 2sC^2 \int_{-s}^{s} \left\| \hat{\psi}(.,t_j,\tau) - \psi(.,t_j,\tau) \right\|^2_{\mathcal{L}^2(\mathcal{X},\rho_{\mathcal{X}})} d\tau \\
&\leq (2sC)^2 \sup_{\tau,j=1...m} \left\| \hat{\psi}(.,t_j,\tau) - \psi(.,t_j,\tau) \right\|^2_{\mathcal{L}^2(\mathcal{X},\rho_{\mathcal{X}})}.
\end{aligned}
$$

Recall that: $\hat{\psi}(x,t,\tau) = \frac{1}{|G|}\sum_{i=1}^{|G|} \mathbb{1}_{\langle g_i t, x \rangle \leq \tau}$, and $\psi(x,t,\tau) = \mathbb{E}_g \hat{\psi}(x,t,\tau)$.
Clearly $\left\| \mathbb{1}_{\langle .,gt \rangle \leq \tau} \right\|_{\mathcal{L}_2(\mathcal{X},\rho_{\mathcal{X}})} \leq 1$, hence applying again Lemma 7, for $\delta_2 > 0$ we have with probability $1 - \delta_2$:

$$\left\| \hat{\psi}(.,t_j,\tau) - \psi(.,t_j,\tau) \right\|^2_{\mathcal{L}^2(\mathcal{X},\rho_{\mathcal{X}})} \leq \frac{1}{|G|}\left(1 + \sqrt{2\log\left(\frac{1}{\delta_2}\right)}\right)^2,$$

It follows that: $\forall j = 1...m, \left\| \hat{f}_j - f_j \right\| \leq \frac{2Cs}{\sqrt{|G|}}\left(1 + \sqrt{2\log\left(\frac{1}{\delta_2}\right)}\right)$, with probability $1 - m\delta_2$.
Hence with probability $1 - m\delta_2$, we have:

$$\left\| \frac{1}{m}\sum_{j=1}^{m}(\hat{f}_j - f_j) \right\|_{\mathcal{L}^2(\mathcal{X},\rho_{\mathcal{X}})} \leq \frac{2Cs}{\sqrt{|G|}}\left(1 + \sqrt{2\log\left(\frac{1}{\delta_2}\right)}\right). \tag{11}$$

and by the approximation of a Riemann sum we have that:

$$\left\| \frac{1}{m}\sum_{j=1}^{m}(\hat{f}_j - \tilde{f}_j) \right\|_{\mathcal{L}^2(\mathcal{X},\rho_{\mathcal{X}})} \leq \frac{2sC}{n}. \tag{12}$$

It is clear that $\tilde{f} = \frac{1}{m}\sum_{j=1}^{m}\tilde{f}_j \in \tilde{\mathcal{F}}$, hence, putting together equations (19),(20), and (21) we finally have:

$$\left\|\frac{1}{m}\sum_{j=1}^{m}\tilde{f}_j - f\right\|_{\mathcal{L}^2(\mathcal{X},\rho_{\mathcal{X}})} \leq \left\|\frac{1}{m}\sum_{j=1}^{m}(\tilde{f}_j - \hat{f}_j)\right\|_{\mathcal{L}^2(\mathcal{X},\rho_{\mathcal{X}})} + \left\|\frac{1}{m}\sum_{j=1}^{m}(\hat{f}_j - f_j)\right\|_{\mathcal{L}^2(\mathcal{X},\rho_{\mathcal{X}})} + \left\|\frac{1}{m}\sum_{j=1}^{m}f_j - f\right\|_{\mathcal{L}^2(\mathcal{X},\rho_{\mathcal{X}})}$$

$$\leq \frac{2sC}{n} + \frac{2Cs}{\sqrt{|G|}}\left(1 + \sqrt{2\log\left(\frac{1}{\delta_2}\right)}\right) + \frac{2sC}{\sqrt{m}}\left(1 + \sqrt{2\log\left(\frac{1}{\delta_1}\right)}\right)$$

with probability $1 - \delta_1 - m\delta_2$. $\qquad\square$

The following Lemma shows how the approximation of functions in $\mathcal{F}_p$, by functions in $\tilde{\mathcal{F}}$, translates to the expected Risk:

**Lemma 4** (Bound on the Approximation Error). *Let $f \in \mathcal{F}_p$, fix $\delta_1, \delta_2 > 0$. There exists a function $\tilde{f} \in \tilde{\mathcal{F}}$, such that:*

$$\mathcal{E}_V(\tilde{f}) \leq \mathcal{E}_V(f) + \frac{2sLC}{\sqrt{m}}\left(1 + \sqrt{2\log\left(\frac{1}{\delta_1}\right)}\right) + L\left(\frac{2sC}{\sqrt{|G|}}\left(1 + \sqrt{2\log\left(\frac{m}{\delta_2}\right)}\right) + \frac{2sC}{n}\right),$$

*with probability at least $1 - \delta_1 - \delta_2$.*

*Proof of Lemma 5.* $\mathcal{E}_V(\tilde{f}) - \mathcal{E}_V(f) \leq \int_{\mathcal{X}}\left|V(y\tilde{f}(x)) - V(yf(x))\right|d\rho_{\mathcal{X}}(x) \leq L\int_{\mathcal{X}}|\tilde{f}(x) - f(x)|d\rho_{\mathcal{X}}(x) \leq L\sqrt{\int_{\mathcal{X}}(\tilde{f}(x) - f(x))^2 d\rho_{\mathcal{X}}(x)} = L\left\|\tilde{f} - f\right\|_{\mathcal{L}^2(\mathcal{X},\rho_{\mathcal{X}})}$, where we used the Lipschitz condition and Jensen inequality. The rest of the proof follows from Lemma 4. $\qquad\square$

The following Lemma gives a bound on the estimation of the expected Risk with finite training samples:

**Lemma 5** (Bound on the Estimation Error). *Fix $\delta > 0$, then*

$$\sup_{f\in\tilde{\mathcal{F}}}\left|\mathcal{E}_V(f) - \hat{\mathcal{E}}_V(f)\right| \leq \frac{1}{\sqrt{N}}\left(4LsC + 2V(0) + LC\sqrt{\frac{1}{2}\log\left(\frac{1}{\delta}\right)}\right),$$

*with probability $1 - \delta$.*

*Proof.* The proof follows from Theorem 5 given in Appendix D. It is sufficient to bound the Rademacher complexity of the class $\tilde{\mathcal{F}}$:

$$\mathcal{R}_N(\tilde{\mathcal{F}}) = \mathbb{E}_{x,\sigma}\left[\sup_{f\in\tilde{\mathcal{F}}}\left|\frac{1}{N}\sum_{i=1}^{N}\sigma_i f(x_i)\right|\right] = \mathbb{E}_{x,\sigma}\left[\sup_{f\in\tilde{\mathcal{F}}}\left|\frac{s}{Nn}\sum_{i=1}^{N}\sigma_i\left(\sum_{j=1}^{m}\sum_{k=-n}^{n}w_{j,k}\hat{\psi}\left(x_i,t_j,\frac{sk}{n}\right)\right)\right|\right]$$

$$= \mathbb{E}_{x,\sigma}\left[\sup_{f\in\tilde{\mathcal{F}}}\left|\frac{s}{Nn}\sum_{j=1}^{m}\sum_{k=-n}^{n}w_{j,k}\sum_{i=1}^{N}\sigma_i\hat{\psi}\left(x_i,t_j,\frac{sk}{n}\right)\right|\right]$$

$$\leq \mathbb{E}_{x,\sigma}\frac{sC}{mNn}\sum_{j=1}^{m}\sum_{k=-n}^{n}\left|\sum_{i=1}^{N}\sigma_i\hat{\psi}\left(x_i,t_j,\frac{sk}{n}\right)\right| \quad \text{By Holder inequality: } \langle a,b\rangle \leq \|a\|_{\infty}\|b\|_1$$

$$\leq \frac{sC}{mNn}\mathbb{E}_x\sum_{j=1}^{m}\sum_{k=-n}^{n}\sqrt{\mathbb{E}_{\sigma}\left(\sum_{i=1}^{N}\sigma_i\hat{\psi}\left(x_i,t_j,\frac{sk}{n}\right)\right)^2} \quad \text{Jensen inequality, concavity of square root}$$

Note that $\mathbb{E}(\sigma_i\sigma_j) = 0$, for $i \neq j$ it follows that:

$\mathbb{E}_{\sigma}\left(\sum_{i=1}^{N}\sigma_i\hat{\psi}\left(x_i,t_j,\frac{sk}{n}\right)\right)^2 = \mathbb{E}_{\sigma}\sum_{i=1}^{N}\sum_{\ell=1}^{N}\sigma_i\sigma_{\ell}\hat{\psi}\left(x_i,t_j,\frac{sk}{n}\right)\hat{\psi}\left(x_{\ell},t_j,\frac{sk}{n}\right) = \sum_{i=1}^{N}\hat{\psi}^2\left(x_i,t_j,\frac{sk}{n}\right) \leq N$, since $\hat{\psi}(.,.,.) \leq 1$. Finally:

$$\mathcal{R}_m(\tilde{\mathcal{F}}) \leq \frac{Cs}{\sqrt{N}}.$$

$\square$

We are now ready to prove Theorem 3:

*Proof of Theorem 3.* Let $f_N^* = \arg\min_{f \in \tilde{\mathcal{F}}} \hat{\mathcal{E}}_V(f)$, $\tilde{f} = \arg\min_{f \in \tilde{\mathcal{F}}} \mathcal{E}_V(f)$, $f_p = \arg\min_{f \in \mathcal{F}_p} \mathcal{E}_V(f)$.

$$\mathcal{E}_V(f_N^*) - \min_{f \in \mathcal{F}_p} \mathcal{E}_V(f) = \underbrace{\left(\mathcal{E}_V(f_N^*) - \mathcal{E}_V(\tilde{f})\right)}_{\text{Statistical Error}} + \underbrace{\left(\mathcal{E}_V(\tilde{f}) - \mathcal{E}_V(f_p)\right)}_{\text{Approximation Error}}$$

The first term is the usual estimation or statistical error than we can bound using Lemma 6, we have:

$$\mathcal{E}_V(f_N^*) - \mathcal{E}_V(\tilde{f}) = \left(\mathcal{E}_V(f_N^*) - \hat{\mathcal{E}}_V(f_N^*)\right) + \underbrace{\left(\hat{\mathcal{E}}_V(f_N^*) - \hat{\mathcal{E}}_V(\tilde{f})\right)}_{\leq 0, \text{by optimality of } f_N^*} + \left(\hat{\mathcal{E}}_V(\tilde{f}) - \mathcal{E}_V(\tilde{f})\right)$$

$$\leq 2 \sup_{f \in \tilde{\mathcal{F}}} \left|\mathcal{E}_V(f) - \hat{\mathcal{E}}_V(f)\right|$$

$$\leq 2 \frac{1}{\sqrt{N}} \left(4LsC + 2V(0) + LC\sqrt{\frac{1}{2}\log\left(\frac{1}{\delta}\right)}\right),$$

with probability $1 - \delta$ over the training samples. Let $\tilde{f}_p$, the function defined in Lemma 4, that approximates $f_p$ in $\tilde{\mathcal{F}}$. By Lemma 5 we know that:

$$\mathcal{E}_V(\tilde{f}_p) \leq \mathcal{E}_V(f_p) + \frac{2sLC}{\sqrt{m}}\left(1 + \sqrt{2\log\left(\frac{1}{\delta_1}\right)}\right) + L\left(\frac{2sC}{\sqrt{|G|}}\left(1 + \sqrt{2\log\left(\frac{m}{\delta_2}\right)}\right) + \frac{2sC}{n}\right),$$

with probability $1 - \delta_1 - \delta_2$, on the choice of the templates and the sampled group elements. By optimality of $\tilde{f} \in \tilde{\mathcal{F}}$, we have

$$\mathcal{E}_V(\tilde{f}) \leq \mathcal{E}_V(\tilde{f}_p) \leq \mathcal{E}_V(f_p) + \frac{2sLC}{\sqrt{m}}\left(1 + \sqrt{2\log\left(\frac{1}{\delta_1}\right)}\right) + L\left(\frac{2sC}{\sqrt{|G|}}\left(1 + \sqrt{2\log\left(\frac{m}{\delta_2}\right)}\right) + \frac{2sC}{n}\right)$$

Hence by a union bound with probability $1 - \delta - \delta_1 - \delta_2$, on the training set , the templates and the group elements we have:

$$\mathcal{E}_V(f_N^*) - \min_{f \in \mathcal{F}_p} \mathcal{E}_V(f) \leq 2\frac{1}{\sqrt{N}}\left(4LsC + 2V(0) + LC\sqrt{\frac{1}{2}\log\left(\frac{1}{\delta}\right)}\right)$$

$$+ \frac{2sLC}{\sqrt{m}}\left(1 + \sqrt{2\log\left(\frac{1}{\delta_1}\right)}\right) + L\left(\frac{2sC}{\sqrt{|G|}}\left(1 + \sqrt{2\log\left(\frac{m}{\delta_2}\right)}\right) + \frac{2sC}{n}\right).$$

$\square$

## C  Technical tools

**Theorem 1.** *[29] Let $X_1, X_2, ..., X_m$ be i.i.d. random variables with cumulative distribution function $F$, and let $\hat{F}_m$ be the associated empirical cumulative density function $\hat{F}_m = \frac{1}{m}\sum_{i=1}^m \mathbb{1}_{X_i \leq \tau}$. Then for any $\gamma > 0$*

$$\mathbb{P}\left\{\sup_\tau \left|\hat{F}_m(\tau) - F(\tau)\right| > \gamma\right\} \leq 2\exp\left(-2m\gamma^2\right).$$

**Lemma 6** ([15],Concentration of the mean of bounded random variables in a Hilbert Space). *Let $(\mathcal{H}, \langle., .\rangle_\mathcal{H})$ be a Hilbert space. Let $X_j$, $j = 1 \ldots K$, be iid random, such that $||X_j||_\mathcal{H} \leq M$. Then for any $\delta > 0$, with probability $1 - \delta$,*

$$\left\|\frac{1}{K}\sum_{j=1}^K X_j - \frac{1}{K}\mathbb{E}\sum_{j=1}^K X_j\right\|_\mathcal{H} \leq \frac{M}{\sqrt{K}}\left(1 + \sqrt{2\log\left(\frac{1}{\delta}\right)}\right).$$

**Theorem 2** ([15]). *Let $\mathcal{F}$ be a bounded class of function, $\sup_{x \in \mathcal{X}} |f(x)| \leq C$ for all $f \in \mathcal{F}$. Let $V$ be an $L$-Lipschitz loss. Then with probability $1 - \delta$, with respect to training samples $\{x_i, y_i\}_{i=1\ldots N}$, every $f$ satisfies:*

$$\mathcal{E}_V(f) \leq \hat{\mathcal{E}}_V(f) + 4L\mathcal{R}_N(\mathcal{F}) + \frac{2V(0)}{\sqrt{N}} + LC\sqrt{\frac{1}{2N}\log\frac{1}{\delta}},$$

*where $\mathcal{R}_N(\mathcal{F})$ is the Rademacher complexity of the class $\mathcal{F}$:*

$$\mathcal{R}_N(\mathcal{F}) = \mathbb{E}_{x,\sigma}\left[\sup_{f \in \mathcal{F}}\left|\frac{1}{N}\sum_{i=1}^{N}\sigma_i f(x_i)\right|\right],$$

*the variables $\sigma_i$ are iid symmetric Bernoulli random variables taking value in $\{-1, 1\}$, with equal probability and are independent form $x_i$.*

## D  Numerical Evaluation

### D.1  Permutation Invariance Experiment

For our first experiment, we created an artificial dataset which was designed to exploit permutation invariance, providing us with a finite group to which we had complete access. The dataset $X_{perm}$ consists of all sequences of length $L = 5$, where each element of the sequence is taken from an alphabet $A$ of 8 characters, giving us a total of 32,768 data points. Two characters $c_1, c_2 \in A$ were randomly chosen and designated as targets, so that a sequence $x \in X_{perm}$ is labeled positive if it contains both $c1$ and $c_2$, where the position of these characters in the sequence does not matter. Likewise, any sequence that does not contain both characters is labeled negative. This provides us with a binary classification problem (positive sequences vs. negative sequences), for which the label is preserved by permutations of the sequence indices, i.e. two sequences will belong to the same orbit if and only if they are permuted versions of one another.

The $i^{\text{th}}$ character in $A$ is encoded as an 8-dimensional vector which is 0 in every position but the $i^{\text{th}}$, where it is 1. Each sequence $x \in X_{perm}$ is formed by concatenating the 5 such vectors representing its characters, resulting in a binary vector of length 40. To build the permutation-invariant representation, we project a binary sequences onto an equal-length sequence consisting of standard-normal gaussian vectors, as well as all of its permutations, and then pool over the projections with a CDF.

As a baseline, we also used a bag-of-words representation, where each $x \in X_{perm}$ was encoded with an 8-dimensional vector with $i^{\text{th}}$ element equal to the count of how many times character $i$ appears in $x$. Note that this representation is also invariant to permutations, and so should share many of the benefits of our feature map.

For all classification results, 4000 points were randomly chosen from $X_{perm}$ to form the training set, with an even split of 2000 positive points and 2000 negative points. The remaining 28,768 points formed the test set.

We know from Theorem 3 that the expected risk is dependent on the number of templates used to encode our data and on the number of bins used in the CDF-pooling step. The right panel of Figure 1 shows RLS classification accuracy on $X_{perm}$ for different numbers of templates and bins. We see that, for a fixed number of templates, increasing the number of bins will improve accuracy, and for a fixed number of bins, adding more templates will improve accuracy. We also know there is a further dependence on the number of transformation samples from the group $G$. The left panel of figure 1 shows how classification accuracy, for a fixed number of training points, bins, and templates, depends on the number of transformation we have access to. We see the curve is rather flat, and there is a very graceful degradation in performance.

In Figure 2, we include the sample complexity plot (for RLS) with the error bars added.

### D.2  TIDIGITS Experiment

Here, we add plots showing performance as a function of number of templates and bins for some other splits of the TIDIGITS data.

Figure 1: *Left) Classification accuracy of random invariant features as function of the number of sampled group elements on $X_{perm}$. Right) Classification accuracy of random invariant features as function of the number of templates and bin sizes on $X_{perm}$.*

Figure 2: *Classification accuracy as a function of training set size. $\Phi = CDF(n, m)$ refers to a random feature map with $n$ bins and $m$ templates. For each training set size, the accuracy is averaged over 100 random training samples. With enough templates/bins, the random feature map outperforms the raw features as well as a bag-of-words representation (also invariant to permutation). We also train an RLS classifier with a haar-invariant kernel, which naturally gives the best performance. However, by increasing the number of templates, we come close to matching this performance with random feature maps.*

Figure 3: *Mean classification accuracy as a function of number of templates, $m$, and bins, $n$. Accuracy is averaged over 30 random template samples for each $m$ and error bars are displayed. In the "Utterance" dataset, we train and test on the same speakers, but the test set contains new utterances of each digit. This is the easiest dataset, representing only intraspeaker variability, and the performance is quite good even for a small number of bins.*

Figure 4: *Mean classification accuracy as a function of number of templates, $m$, and bins, $n$. Accuracy is averaged over 30 random template samples for each $m$ and error bars are displayed. In the "Age (Women)" dataset, we train on adult women and test on children, giving us an age mismatch. Despite this mismatch, performance remains strong.*