[Reviews · NeurIPS 2015]

Submitted by Assigned_Reviewer_1

Haar-group invariant kernels have the form:

K(x,z) = \int_G \int_G k(gx, g'z) dm(g) dm(g'), (1)

where G is a compact unitary group. Therefore, we say these kernels are group invariant since k(x,z) = k(gx,g'z) for any pair of transformations (g,g') belonging to the group G. For example, if we deal with images, an useful group G could a collection of translations and rotations.

Since the double integral in (1) is expensive to compute, the authors propose to use sampling to compute a finite dimensional, randomized feature map that approximates the kernel K. In particular, the feature map is constructed by sampling random elements g from G using the measure m(g), sampling "templates" t from a reference distribution (which is set to be Gaussian in this paper), and then constructing the discretized CDF of the random variable $\langle gt,x \rangle$, for an input x.

The construction is accompanied with concentration inequalities that depict how fast does this randomized feature map converge to the true Haar-group-invariant kernels, and an adaptation of the risk bounds from [Rahimi and Recht, 2007] for the introduced group-invariant feature map.

Question: how does the number of templates affect the quality of approximation? It seems like all asymptotic guarantees are wrt n (the number of bins in the CDF discretization) and not wrt m (the number of templates, as played with in the experiments). What is the learning rate between K_s and K?

The numerical simulations seem preliminary, as the method is demonstrated only on two small-scale datasets. Furthermore, there is no comparison to other related methods (such as regular random features where the number of them is increased at no cost thanks to, for example, a Fastfood approximation). Also, the plots are difficult to interpret: perhaps a logarithmic scale on the Y-axis would help.

Question: how would you learn G if you are not sure about the nature of your data? This approach seems to invoke the human to be more in the loop, drifting one step away from automatic machine learning.

The technical quality of the paper is good, and it is fairly well written. The significance of the paper is undermined by the minimal experimental section, which is conducted on small-scale digit datasets (this Section would be greatly strengthened by incorporating competing methods such as large random feature ensembles, histogram random features, convolutional neural networks... And extending the simulations to larger datasets such as natural images ).

Summary: A technique to efficiently compute randomized feature maps that approximate Haar-group-invariant kernels.

Submitted by Assigned_Reviewer_2

There are two separate ideas in this paper: a new proposal for an invariant kernel, and a random feature approximation to it (though developed in reverse order).

The random features involve computation of empirical distribution of dot products between the input and transformed random templates. The paper shows that the implied inner product is an estimator for the proposed kernel which is shown to be related to the mean distance between transformation orbits. Some generalization theory is also developed around sample complexity of invariant learning. Finally, limited empirical results show some promise with the proposed approach.

A basic comparison that is lacking is how the methods perform relative to convolution neural networks, and other invariant kernel approaches proposed in the literature. For example, one can do invariant learning through the virtual samples technique, which can also be scaled up using the random features approach.

Is the choice of Gaussian distribution for templates mainly for analysis reasons? Or can the templates also be learnt?

Generally, in kernel approximation literature, the Gram matrix approximation is studied together with final downstream accuracy although the latter is more important in practice. But as a first step, it would have been interesting to see how the number of bins (discretization of threshold) and the number of templates affects Gram matrix reconstruction.

What about the case where some of the transformations acting on a domain are unknown, i.e., can unknown transformation groups be estimated as part of the learning process?

Summary: An interesting and novel approach to construct invariant random features based on approximations to Haar integration kernels. Limited, but somewhat promising experimental results.

Submitted by Assigned_Reviewer_3

This paper provide a kernel perspective of the i-theory. The authors provide random features for Haar integration kernel, which is invariant to a group of operators. The authors derive the closed form of the random features and reveal the connection between the kernel to distance in orbits. Upper bounds for kernel approximation and function learning with the random features are derived.

The paper is well-organized and easy to follow. However, I have several concerns listed below.

1), how to select $g$ function in the group is not clear discussed. It will be better to provide some guideline or examples about the design of the group of operators and the measure conducted on the group. Moreover, if the operators are rotations, shifting and so on, the connections between the proposed algorithm and virtue samples is an important point to discuss.

2), in theorem 1, the connection between the closed form of Haar kernel and the distance between orbits

highly depends on the template measure and difficult to generalize to other template distribution. In experiments, the authors did not use the truncate Gaussian distribution. It will be better if the theorem 1 is justified empirically in experiment part.

3), in experiment part, the comparison between the

Haar kernel and the proposed random features, e.g., $\|K - \Phi\top\Phi\|$, will be more directly to justify the theorem 1. Moreover, two important competitors, i.e., using virtue sampling in training procedure for traditional kernel machines, and kernel machine with Haar kernel, should be involved into the experiment.

There are several typos in the paper.

i), the definition of $\perp$ in the template measure in line 153 is not clearly defined. ii), in line 138, the number of equations are not correct.
Summary: The authors provide a kernel view of i-theory. The Haar kernel and its corresponding random features are discussed. The theoretical analysis and empirical comparison of the kernel approximation error and function learning error using the proposed random features are provided.

Author Feedback
Author rebuttal: We thank the reviewers for their constructive and encouraging comments and address in the following their concerns.
*Reviewer 1*: Thanks for the encouraging comments.
*Reviewer 2*: 1- Convnet is indeed an important baseline for translation invariance. One can see the random feature map with the translation group as implementing a convolution (actually a correlation) with a CDF pooling. Learning with virtual examples is also a way to learn invariance, but it is not yet clear theoretically how this compares to learning with an invariant representation. The framework introduced in this paper allows such study and we are currently working on it. We were interested in this paper in learning from few examples, using the invariance prior.
2-The choice of gaussian templates is needed for technical reasons. The templates can be learned in an unsupervised way. With this kernel approximation view, we envision two ways for learning the templates in an unsupervised way. The first a la Nystrom in approximating the kernel matrix, the second by finding the optimal templates that minimize the distortion of the embedding in Theorem 2.
3 -The kernel approximation error in the operator norm can be analyzed as well, thanks for pointing this out. Sharper generalization analysis can be conducted using this [18]. We will include this in a longer version of this paper.
4- For unknown transforms, using sequences of transforming templates (videos of unknown transforms) to build an invariant kernel is a promising venue, used usually in the slow subspace learning, and known as the trace rule in neuroscience. We will study this in future work.
*Reviewer 3*:1 - The group elements were selected uniformly at random for the permutation case (full group). Error bars show the robustness to the choice of group elements (figure in the supplemental). For image and speech applications, local invariance is needed, and group elements where fixed in a small subset of the group. We discuss in Remark 4 that this builds partial invariance.
2- Please see points 1 and 2 to reviewer 2 regarding the gaussian sampling, and the virtual examples. We will include a plot on permutations justifying the kernel approximation in Theorem 1.
3- For the permutation example, we included ridge regression with the Haar kernel (figure 1 left). For other experiments this method does not scale.
4- Thanks for pointing the typos.
*Reviewer 4*: 1- The approximation error in the RKHS depends on m as O(1/sqrt{m}). For approximation the kernel matrix itself we will include a result in a longer version (Please see point 3 Reviewer 2 ).
2 -Thanks for the comments regarding the plots we will enhance the visualization of the plots.
3- Regarding learning the templates and the groups, please see points 2 and 3 for Reviewer 2.
4- Random features by themselves cannot generalize in a low sample complexity scenario, unless we allow virtual examples. We focused in this paper on a random feature map with built in invariance, and proved invariance at the function space level from a low sample complexity. It would be interesting to analyze if random features with virtual examples induce invariance at the function space level, but this does not seem to be trivial to prove.
*Reviewer 5*: The use of gaussian truncation is preferred to the uniform one because we rely heavily on the rotation invariance of gaussian, as well as on bivariate gaussian correllation in our analysis. Unfortunately, a concentration result shows that this is true for uniform sampling, we will still pay the epsilon price (K_Uniform close to K_Gaussian that is close to K).
*Reviewer 6*: Thanks for the comment, we will enhance the previous work section.